# Fetal programming through early weaning shapes the metabotype of Nelore heifers

Anjaleena Yaseen[1,2], Ana Laura dos Santos Munhoz Gôngora[3], Thiago Kan Nishimura[4], Gabriel Henrique Ribeiro[5,6], Eduardo Solano Pina Santos[2], Alanne Tenório Nunes[7], Tarique Hussain[1], Amjad Hameed[1], Luiz Alberto Colnago[5], Paulo Roberto Leme[2], Arlindo Saran Netto[2], Guilherme Pugliesi[4], Nara Regina Brandão Cônsolo[3]*

1 Animal Sciences Division, Nuclear Institute for Agriculture and Biology College, Pakistan Institute of Engineering and Applied Sciences, Faisalabad, Pakistan, 2 Department of Animal Science, College of Animal Science and Food Engineering, University of São Paulo, Pirassununga, São Paulo, Brazil, 3 Department of Nutrition and Animal Production, School of Veterinary Medicine and Animal Science, University of São Paulo, Pirassununga, São Paulo, Brazil, 4 Department of Animal Reproduction, School of Veterinary Medicine and Animal Science, University of São Paulo, Pirassununga, São Paulo, Brazil, 5 Embrapa Instrumentação, São Carlos, Brazil, 6 Department of Chemistry, Federal University of São Carlos" , São Carlos, São Paulo, Brazil, 7 Department of Veterinary Medicine, College of Animal Science and Food Engineering, University of São Paulo, Pirassununga, São Paulo, Brazil

* nara.consolo@usp.br

## Abstract

This study investigated the effects of fetal programming induced by early weaning (EW) of the previous calf on the serum metabotype of Nelore heifers. A 2 × 2 factorial design was employed, considering maternal category (multiparous vs. second-parity) and weaning strategy [EW vs. conventional weaning (CW)]. A total of 55 heifers were evaluated, including 25 born to cows subjected to EW (15 multiparous, 10 second-parity) and 30 born to cows subjected to CW (15 multiparous, 15 second-parity). Serum samples were collected from heifers at an average age of 14 months prior to the initiation pf the fixed-time artificial insemination (FTAI) protocol, and metabolomic profiling was conducted using ¹H-NMR spectroscopy. Data analysis was performed using R software and MetaboAnalyst 6.0. The effects of weaning strategy and maternal category were assessed through partial least squares-discriminant analysis (PLS-DA) and differential abundance analysis, while enrichment analysis was applied to identify biological pathways associated with the observed metabolic changes. The results demonstrated significant effects of weaning strategy, maternal category, and their interaction on the serum metabotype. Heifers born to second-parity cows subjected to EW exhibited elevated concentrations of glutamate, acetate, citrate, and hippurate compared to those subjected to CW, whereas no significant differences between weaning strategies were observed in heifers born to multiparous cows. Weaning strategies modulated metabolic pathways related to energy metabolism, amino acid utilization, and gluconeogenesis. Furthermore, heifers born to multiparous cows displayed higher serum levels of phenylalanine, creatine phosphate,

**Data availability statement:** All relevant data are within the paper and its Supporting Information files.

**Funding:** This study was supported by The Coordenação de Aperfeiçoamento de Pessoal de Nível Superior - Brasil (CAPES) - Finance Codes 001, and Fundação de Amparo à Pesquisa do Estado de São Paulo (FAPESP) grant 2020/08845-3, 2017/18937-0, 2022/07906-4 and 2025/00509-8. The funders had no role in study design, data collection and analysis, decision to publish, or preparation of the manuscript.

**Competing interests:** The authors have declared that no competing interests exist.

and glutamine, indicating enhanced nutrient availability for growth, development, and metabolic programming compared to their counterparts born to second-parity cows. These findings underscore the potential of EW to improve metabolic efficiency and nutrient reallocation, particularly under conditions of limited feed availability or heightened physiological demands. Overall, this study offers valuable insights into the metabolic programming of beef cattle, supporting strategies to optimize weaning protocols and enhance productivity in cattle production systems.

## 1. Introduction

Fetal developmental programming refers to the intricate interplay between fetal growth and long-term physiological outcomes. [1]. This concept posits that the maternal environment during critical windows of development can profoundly and permanently shape an individual's physiology and metabolism. The processes of prenatal organogenesis and tissue differentiation are tightly modulated by maternal nutrition, metabolic status, and environmental factors. [2,3]. In response to these maternal cues, the fetus undergoes adaptive cellular, biochemical, and molecular changes, largely mediated by placental function and cellular metabolism. [4,5]. These adaptations not only influence prenatal growth and maturation but also exert long-lasting effects on reproductive potential, skeletal muscle development, meat quality, and postnatal physiological functions. [6,7].

Beef cattle production in Brazil's tropical regions predominantly relies on *Bos indicus* breeds and faces substantial challenges during the dry season due to a marked decline in forage quality and availability [8]. Prolonged droughts, particularly in late summer and autumn, lead feed shortages and low-quality forage that fails to meet the nutritional demands of pregnant beef cows and their developing fetuses [8,9]. Furthermore, since weaning typically occurs at approximately 8 months of age in *Bos indicus* cows [10], lactating cows experience elevated energy and nutrient requirements to sustain milk production during the period of rapid fetal growth [11]. Notably, primiparous cows have even higher energy demands than multiparous cows, as they require additional nutrients to support their own continued growth [12]. This nutritional stress can detrimentally affect fetal development, exerting long-lasting impacts on the offspring [13]. When these increased demands are not met, a competition for nutrients emerges between milk production and fetal development, potentially compromising the offspring's reproductive growth and pubertal development [7,14,15].

In this context, advancing weaning from 8 to 5 months in *Bos indicus* females can positively impact pregnancy outcomes by reducing the nutritional demands of lactating cows during the dry season. This practice enables the reallocation of nutrients toward fetal growth and development, fostering long-term improvements in offspring health, metabolism, productivity, and reproductive performance, thereby enhancing the sustainability of beef cattle production systems in tropical and subtropical regions [16–18]. Accordingly, elucidating the metabolic profiles of programmed animals can

provide critical insights into how fetal programming induced by early weaning of the previous calf influences or differentiates offspring phenotypes.

Metabolic imprinting, shaped by genetics, epigenetics, and environmental factors, plays a pivotal role in determining an individual's metabolic profile, or metabotype [19]. This process affects critical aspects of animal physiology, including genetic potential, the onset of puberty, and overall productivity and reproductive performance [20–22]. Early weaning, within the framework of fetal programming, offers a valuable opportunity to explore the metabolic pathways that govern the long-term metabolic state and reproductive development of Nelore heifers [23]. Such insights can inform the design of weaning strategies, nutritional interventions, and management practices aimed at enhancing herd health and productivity. Nevertheless, the interplay between fetal programming, the age at weaning of the previous calf, and the metabolic processes associated with puberty in heifers remains largely unexplored.

We hypothesized that early weaning of the previous calf may induce alterations in the metabolic profile of heifers, potentially differing according to maternal parity. To test this hypothesis, a metabolomics approach was employed to investigate the impact of fetal programming through early weaning on the metabotype of heifers prior to FTAI. By analyzing the metabotype, this study aims to provide deeper insights into how early weaning influences fetal programming, particularly regarding the underlying metabolic mechanisms, and whether variations exist in the heifers' metabotype according to the parity order of the cow. These findings may inform the development of more effective management strategies to improve reproductive performance and overall productivity in beef cattle systems.

## 2. Methodology

### 2.1. Ethics statement

The experiment protocol was reviewed and approved by the Ethics Committee of the School of Animal Science and Food Engineering, University of São Paulo, under the protocol number 2884250620. All procedures were conducted at the University of São Paulo, Pirassununga campus (São Paulo, Brazil).

### 2.2. Experimental design, animals and management

This study employed a 2×2 factorial design with two main factors: parity (second-parity or multiparous) and weaning strategy (early weaning, EW, at 5 months of age; or conventional weaning, CW, at 8 months age). A total of 208 cows (57 primiparous and 151 multiparous at the beginning of the experiment) were included. These cows calved in 2019 and became pregnant via FTAI during the 2020 breeding season (November 2020 to January 2021). Animals were randomly assigned to one of two weaning strategies: 1) EW, where calves were weaned at approximately 150 days of age (mean±SEM: 149±2.0 days), including 27 primiparous and 74 multiparous cows; and 2) CW, where calves were weaned at approximately 240 days of age (mean±SEM: 247±2.4 days), including 27 primiparous and 77 multiparous cows. During the 2021 calving season (September to October), a total of 55 female Nellore heifers were included in this study. Of these, 25 calves were born to cows subjected to early weaning (15 from multiparous cows and 10 from second-parity cows), and 30 calves were born to cows subjected to conventional weaning (15 from multiparous cows and 15 from second-parity cows). The calves were kept with the cows on *Brachiaria spp*. pasture, with unrestricted access to water, until weaning at 240 days of age. Animals management and diets followed protocols described by Nishimura et al. [24–26].

### 2.3. Blood sample collection

Blood samples (approximately 10 mL) were collected via jugular venipuncture using evacuated tubes (BD Vacutainer, São Paulo, Brazil) without anticoagulant, immediately prior to the initiation of the FTAI protocol, when the heifers were approximately 14 months of old (13.5±0.11 months). Samples were centrifuged at 2,800×g for 15 minutes at 4°C. The resulting serum was transferred into labeled plastic tubes and stored at −80°C until analysis.

## 2.4. Metabolites extraction

A double extraction protocol was performed to isolate serum metabolites for NMR analysis, as described by Da Costa et al. [27]. Initially, 500 μL of serum were transferred into 1.5-mL conical tubes, and 400 μL of ice-cold methanol and 400 μL of ice-cold acetone were added, followed by vortexing for 10 seconds. Then, 400 μL of ice-cold chloroform was added, and the samples were vortexed again for 10 seconds. The samples were placed on ice for 10 minutes before centrifuged at 10,000 × g for 10 minutes at 4°C to precipitate proteins. This extraction process was repeated to enhance metabolite recovery. For the second extraction, approximately 550 μL of the supernatant was transferred to a new 2 mL Eppendorf tube. Ice-cold methanol (300 μL) and acetone (300 μL) were added, followed by vortexing for 10 seconds. Next, 300 μL of ice-cold chloroform was added, the samples were vortexed again for 10 seconds. After 10 minutes on ice, the samples underwent centrifugation at 10,000 × g for 10 minutes at 4°C to remove residual proteins. Finally, approximately 1 mL of the supernatant was transferred to a new 2 mL conical tube for pooling and subsequent analysis.

## 2.5. NMR spectroscopy and spectral processing

**Sample preparation for NMR measurements.** The freeze-dried residues were solubilized in 550 μL of phosphate buffer ($D_2O$-based PBS; 0.1 mM; pD = 7.4) containing 0.5 mM of sodium 3-(Trimethylsilyl)-1-propanesulfonic acid (DSS, from Cambridge Isotopes, Leicestershire, UK), used as the internal standard for NMR. Subsequently, 550 μL of each sample was transferred to standard 5 mm NMR tube for NMR measurements.

**NMR spectroscopy.** One-dimensional (1D) $^1$H-NMR spectroscopy was utilized for metabolite profiling, with analyses conducted at EMBRAPA Instrumentation (São Carlos, São Paulo, Brazil), as described by Cônsolo et al. [20]. Briefly, $^1$H NMR spectra were acquired on a 14.1 T (600 MHz for hydrogen frequency) Bruker spectrometer (Bruker Corporation, Karlsruhe, Baden-Wurttemberg, Germany), model Avance III HD, equipped with a BBO 5-mm probe, at a temperature of 298.15K. All $^1$H NMR spectra were acquired using a pulse sequence with H2O presaturation, named by Bruker as noesygppr1d pulse sequence, with the following acquisition parameters: 90° pulse duration of 12.59 ms, recycle delay of 25 s, 106k data points, 128 scans, acquisition time of 4.51 s, spectral width of 20.02 ppm, 50 ms mixing time (d8), and 4 dummy scans. The FIDs of the spectra are available on Supplementary S1 Table.

## 2.6. Metabolite identification and quantification

The $^1$H spectra were processed with a 0.3 Hz line broadening using TopSpin™ 3.6.1 software (Bruker, Biospin, Germany). Phase and baseline correction were performed using Chenomix Processor NMR Suite Professional 7.7 software (Chenomix Inc, Edmonton, Canada). Metabolites were identified in the $^1$H NMR spectra using the built-in compound library within the Chenomx Profiler tool, where overlapping signals of compounds were also deconvoluted by the software. Metabolite assignments were further confirmed by analyzing 2D NMR correlation spectra using MestRenova software and the Human Metabolome Database.

Also, using the Chenomx Profiler software, all identified metabolites were quantified in each 1H NMR spectrum by integrating the signal areas with the software. The concentration of metabolites present in the sample was determined using the internal standard DSS (1.0 mM, signal at 0.00 ppm). Quantification was performed by comparing the selected metabolite signal to the area of the DSS signal, which had a known concentration of 0.5 mM in all samples. The metabolite concentration data was transferred to Excel for Statistical analyses.

## 2.7. Statistical analysis

Statistical analyses were conducted using the R software. First, normality of residuals and the homogeneity of variances were verified using the UNIVARIATE procedure. Metabolite data were analyzed using a completely randomized design with the MIXED procedure, considering the fixed effect of dam parity order (multiparous and second-parity), weaning

strategy (EW or CW), and their interaction, with the animal as the experimental unit. Means were compared using Student's t-test. Significance was declared at $p < 0.05$.

The metabolomic data were also analyzed using MetaboAnalyst 6.0 (http://www.metaboanalyst.ca/). Before analysis, data were preprocessing, including log transformation and Pareto scaling. Differential abundance analysis was conducted to identify metabolites with significant differences between groups, with p-values adjusted using false discovery rate (FDR) correction. Volcano plots were then generated to visualize these results. Metabolites with FDR-adjusted p-values $< 0.05$ were considered significant. Enrichment analysis was performed on the significant metabolites, using a threshold of $p < 0.1$, to identify biological processes or pathways associated with the observed metabolic changes.

## 3. Results

### 3.1. Serum metabolites profile

Thirty-nine metabolites were quantified in the $^1$H-NMR spectrum of blood serum. The identified metabolites included non-essential amino acids, essential amino acids, biogenic amines, organic acids, sugars, and vitamins (Supplementary S2 Table). The influence of maternal category, weaning strategies, and their interaction on metabolite concentrations in serum samples was assessed. Metabolites that exhibited significant differences among groups are summarized in Table 1. A significant interaction between parity and weaning strategies was observed for glutamate ($p < 0.01$), as well as for acetate, citrate, and hippurate ($p < 0.05$). Specifically, the early weaning (EW) strategy was associated with increased concentrations of these metabolites compared to conventional weaning (CW) in second-parity cows, whereas no significant effect on metabolite concentrations was observed between the weaning strategies in multiparous cows.

PLS-DA analysis was used to visualize the differences in metabolite profiles between the EW and CW treatments, and between the maternal categories (multiparous vs. second-parity). The component 1/component 2 scores plot (Figs 1 and 2) shows overlap between the CW and EW treatments, as well as between the parity categories.

To further investigate the influence of weaning strategies and maternal category on the serum metabolome profile of heifers, differential expression analysis was performed using a t-test, with P-values adjusted by false discovery rate (FDR) correction. The results were visualized using volcano plots (Figs 3 and 4). Regarding weaning strategy, the EW group exhibited significantly higher concentrations ($p < 0.05$) of valine, alanine, lactate, methionine, arginine, hippurate, isoleucine, and creatine phosphate compared to the CW group (Fig 3). Concerning maternal category, heifers born to multiparous cows demonstrated a metabolome characterized by higher concentrations of phenylalanine, creatine phosphate, glutamine, glutamate, and 3-hydroxyisovalerate compared to those born to second-parity cows (Fig 4).

**Table 1. Effects of maternal category order and weaning strategy on metabolite concentration (mmol) in serum samples of Nelore (*Bos indicus*) heifers at 14 months of old.**

| Metabolites (mmol) | Multiparous | | Second-parity | | P-value |
|---|---|---|---|---|---|
| | EW[1] | CW[2] | EW[1] | CW[2] | PAR*WE[3] |
| Acetate | 0.5016[a] | 0.608[a] | 0.6617[a] | 0.3898[b] | 0.026* |
| Citrate | 0.0658[a] | 0.0719[a] | 0.089[a] | 0.0534[b] | 0.029* |
| Glutamate | 0.1507[ab] | 0.1804[a] | 0.2143[a] | 0.1034[b] | 0.009** |
| Hippurate | 0.0353[ab] | 0.034[ab] | 0.0464[a] | 0.022[b] | 0.045* |

[1]Earling weaning; [2]Conventional weaning; [3]Interaction between parity and weaning strategy.

[a,b]Means differing statistically.

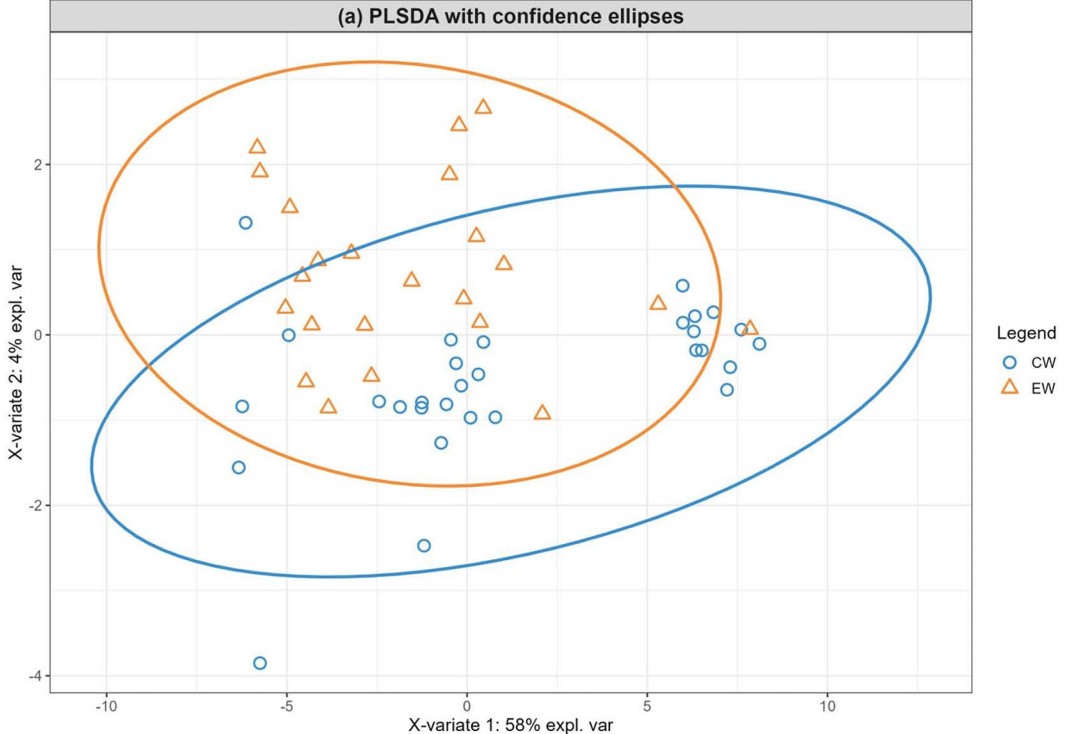

**Fig 1. Partial least-squares discriminant analysis (PLS-DA) of serum metabolite distribution according to conventional weaning (CW, at 8 months age) and early weaning (EW, at 5 months age).** There were two significant Components which together explain 62% of the total variance (Component 1 58% and Component 2 4%). The distribution data presented a partial overlap in the serum metabolome data according to the treatments, suggesting differences in serum metabolome for heifers according to weaning strategies.

### 3.2. Enrichment analysis

For the pairwise comparison of weaning strategies, significant differences ($p < 0.05$) were observed in multiple metabolic pathways, including selenoamino acid metabolism, urea cycle, arginine and proline metabolism, valine, leucine, and isoleucine degradation, spermidine and spermine biosynthesis, glycine and serine metabolism, propanoate metabolism, pyruvate metabolism, glutathione metabolism, glutamate metabolism, alanine metabolism, gluconeogenesis, tryptophan metabolism, and betaine metabolism (Fig 5). These findings highlight the diverse metabolic alterations induced by early weaning compared to conventional weaning. In the comparison of heifers from multiparous and second-parity cows, the most relevant pathways ($p < 0.1$) were nicotinate and nicotinamide metabolism, phenylalanine and tyrosine metabolism, pyrimidine metabolism, lysine degradation, arachidonic acid metabolism, and malate-aspartate shuttle (Fig 6). These results suggest that the metabotype of heifers is influenced by the maternal category, potentially reflecting differences in nutrient allocation and developmental processes.

### 4. Discussion

This study hypothesized that the metabotype of heifers is influenced by fetal programming, triggered by the early weaning (at 5 months age) of the previous calf. The results confirm this hypothesis, revealing significant differences in the metabotypes of heifers from different weaning strategy groups at 14 months of age. Additionally, the metabotype varied based on maternal category, and an interaction between parity and weaning strategy was observed, with the most pronounced differences between the weaning strategies occurred in heifers from second-parity cows.

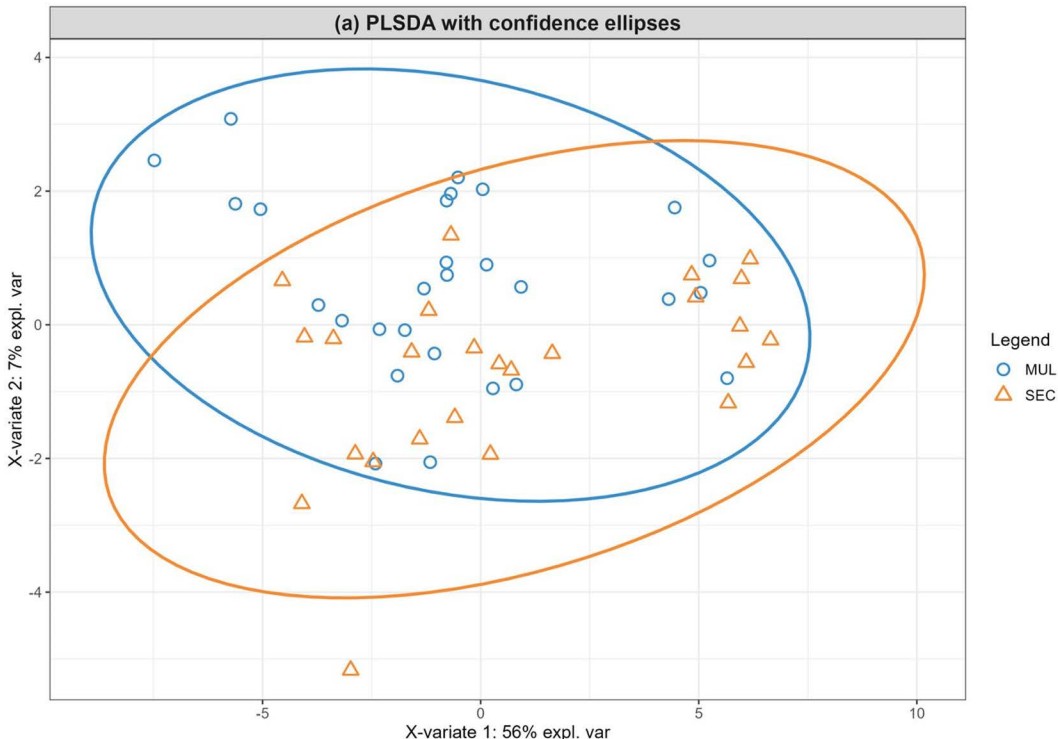

**Fig 2. Partial least-squares discriminant analysis (PLS-DA) of serum metabolite distribution according to maternal category, as multiparous (MUL) and second-parity (SEC).** There were two significant Components which together explain 63% of the total variance (Component 1 56% and Component 2 7%). The distribution data presented a partial overlapping in the serum metabolome data according to the categories, suggesting differences in serum metabolome for heifers according to the dam's parity order.

In heifers born to second-parity cows, the EW strategy was associated with increased concentrations of key metabolites, including glutamate, acetate, citrate, and hippurate, compared to CW. In contrast, weaning strategies did not affect these metabolites in the blood of heifers born to multiparous cows. This disparity may be attributed to the different physiological requirements and adaptive capacities of the two groups. Multiparous cows, having completed their body growth, can efficiently allocate nutrients to fetal development, whereas second-parity cows may still face the dual challenge of supporting their own continued growth alongside fetal nutrient demands [28,29]. Consequently, heifers from second-parity cows may benefit more from the early weaning of the previous calf compared to CW, due to the energetic status of the mothers.

These data are corroborated by Nishimura et al. [24], who evaluated the same animals and reported that early weaning significantly improved the body condition score, body weight, average daily gain, and fat deposition in the carcasses of the cows during the 90 days they were without calves. The authors attribute these results to a reduction in the maternal metabolic demands due to the cessation of lactation, which positively affects the maternal metabolic and nutritional status [24]. This effect is particularly beneficial for second-parity cows, as they are still in a developmental stage.

The elevated concentrations of metabolites observed in heifers from second-parity cows under EW, including glutamate, acetate, citrate, and hippurate, underscore critical metabolic adaptations related to biochemical pathways regulating pubertal onset and reproductive function. Glutamate plays a dual role, participating in protein synthesis as a vital energy substrate for rapidly proliferating tissues [30], and functioning as a principal excitatory neurotransmitter and precursor to GABA [31]. Glutamate modulates GnRH neuron activation within the hypothalamus, potentially advancing the timing of the

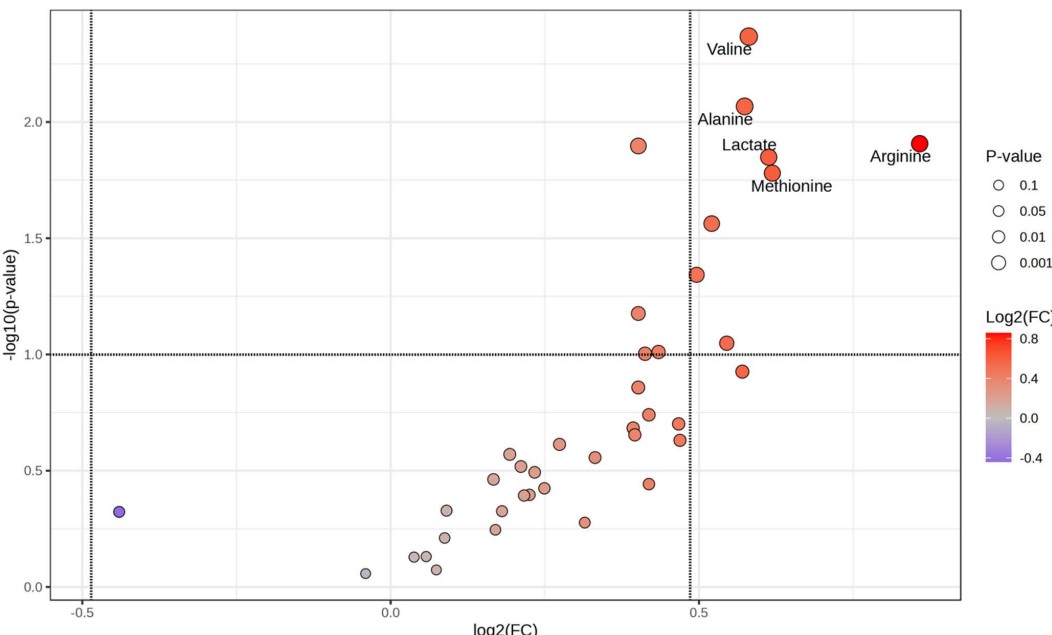

**Fig 3. Volcano plot illustrating the differential expression of serum metabolites in heifers subjected to early weaning (EW, at 5 months age) compared to conventional weaning (CW, at 8 months age) strategies.** Metabolites with significantly higher concentrations in the early weaning (EW) group (p<0.05) include valine, alanine, lactate, methionine, arginine, hippurate, isoleucine, and creatine phosphate.

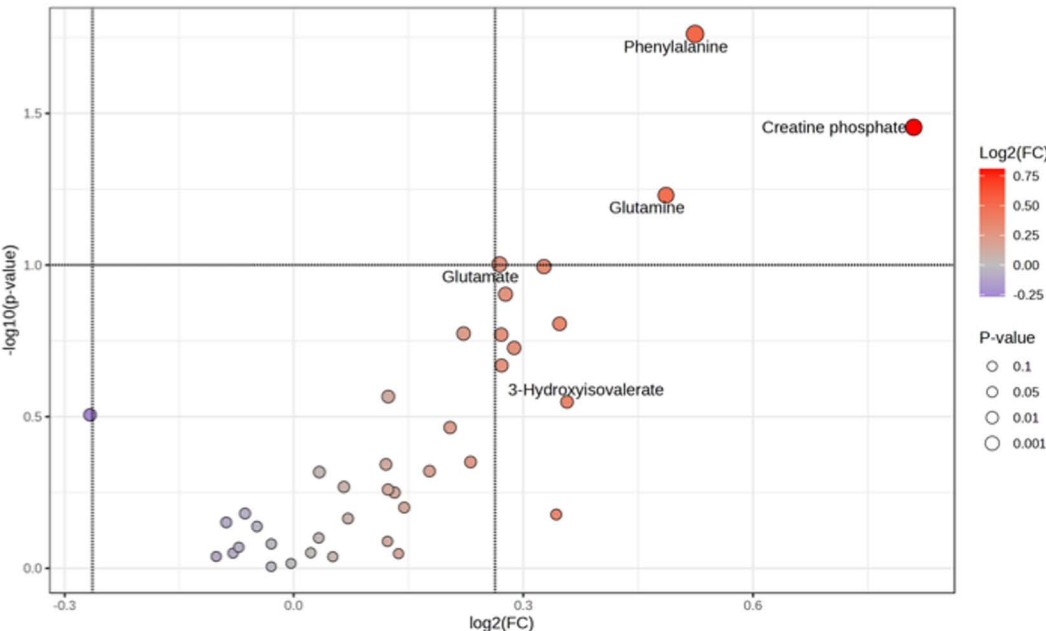

**Fig 4. Volcano plot showing the differential expression of serum metabolites in heifers born to multiparous cows compared to those born to second-parity cows.** Metabolites with significantly higher concentrations in heifers from multiparous cows (p<0.05) include phenylalanine, creatine phosphate, glutamine, glutamate, and 3-hydroxyisovalerate.

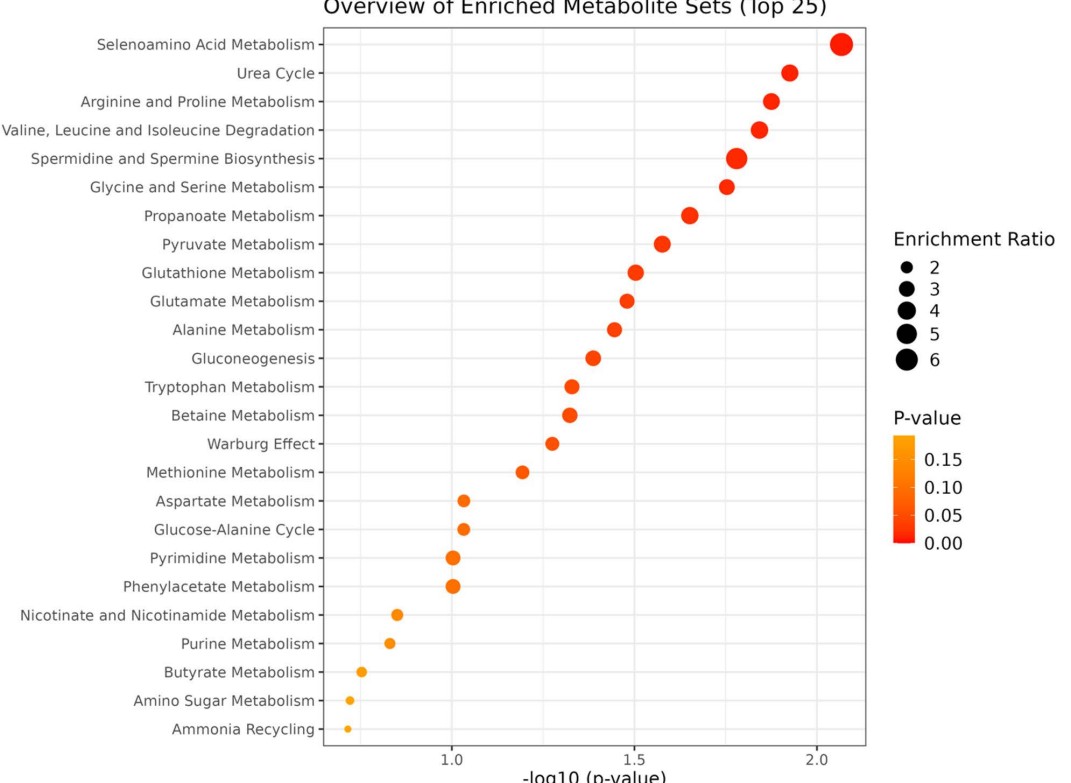

**Fig 5. Enrichment analysis of metabolite sets comparing early and conventional weaning strategies.** Significant pathways (p < 0.05) include selenoamino acid metabolism, urea cycle, arginine and proline metabolism, valine, leucine, and isoleucine degradation, spermidine and spermine biosynthesis, glycine and serine metabolism, propanoate metabolism, pyruvate metabolism, glutathione metabolism, glutamate metabolism, alanine metabolism, gluconeogenesis, tryptophan metabolism, and betaine metabolism.

initial LH release and thereby accelerating the onset of first estrus [32]. Acetate, a primary substrate for lipogenesis [33], promotes adipose tissue accumulation necessary to reach the metabolic threshold for pubertal initiation and contributes to optimizing oocyte metabolism, quality, and early embryonic development [34–36]. Citrate, an intermediate in the tricarboxylic acid cycle, facilitates cellular energy production crucial for growth and metabolic adaptation [37], and has been linked to enhanced mitochondrial metabolic flux and increased ovarian steroidogenesis, emphasizing its role in steroid hormone biosynthesis during reproductive maturation [38]. Finally, hippurate, a glycine conjugate of benzoic acid derived from microbial metabolism of dietary polyphenols, reflects shifts in nutrient absorption and energy partitioning [39] and serves as a biomarker of metabolic health and hepatic homeostasis [40,41], with potential implications for improving immune competence in growing heifers.

The observed differences suggest that EW may heighten metabolic demands in second-parity cows, leading to increased mobilization of these metabolites to sustain both maternal and fetal requirements, as previously discussed and supported by Nishimura et al. [24]. This aligns with findings that heifers are more susceptible to metabolic stress and negative energy balance (NEB) during critical periods [42–44]. Primiparous cows, for example, require greater energy for simultaneous growth and reproductive functions, which can prolong recovery from NEB and delay ovulation [42,43]. Additionally, differences in feed intake, endocrine responses, and blood metabolite levels between primiparous and multiparous cows during transition periods further emphasize their differing metabolic challenges [29,45]. Importantly, the lack of differences in metabolite concentrations between the weaning strategies in multiparous cows suggests that these animals

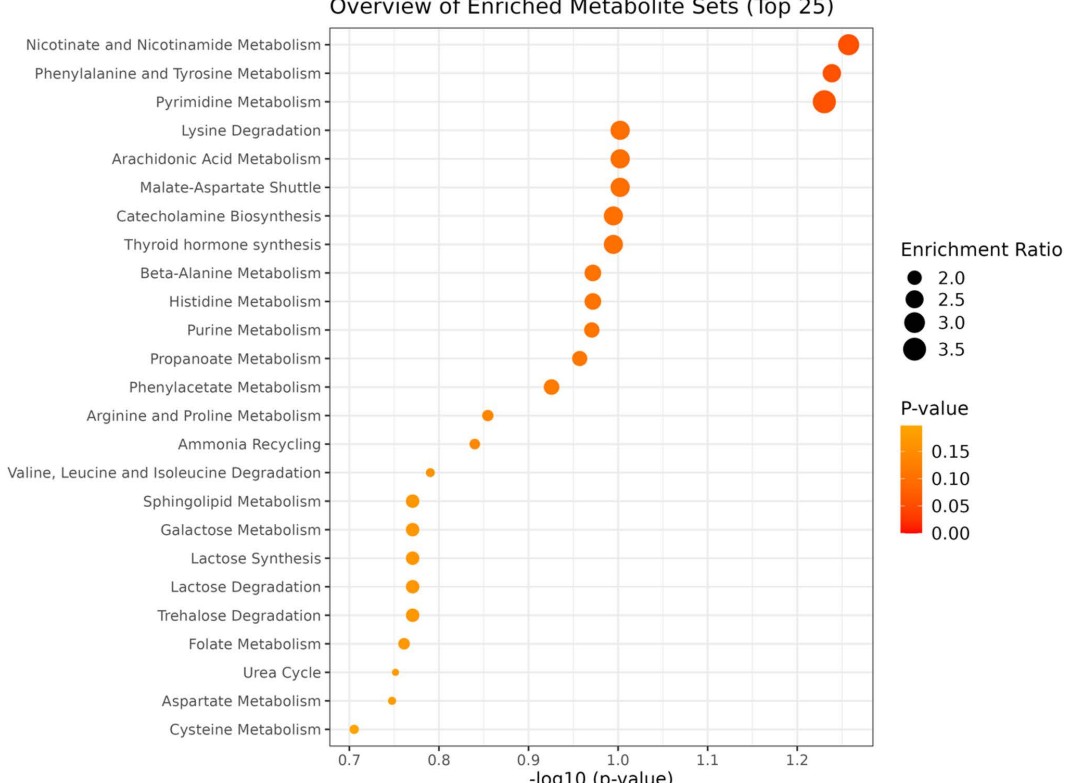

**Fig 6. Enrichment analysis of metabolite sets comparing heifers from multiparous and second-parity cows.** Significant pathways (p < 0.1) include nicotinate and nicotinamide metabolism, phenylalanine and tyrosine metabolism, pyrimidine metabolism, lysine degradation, arachidonic acid metabolism, and malate-aspartate shuttle.

have a greater ability to maintain metabolic stability, likely due to their more mature physiological status and reduced growth-related energy demands. These findings underscore that the impact of early weaning on metabolic programming is particularly significant in younger, still-developing cows.

Furthermore, a comparison of heifers born to multiparous and second-parity cows revealed notable differences in metabotype. Heifers from multiparous cows exhibited higher concentrations of metabolites such as phenylalanine, creatine phosphate, glutamine, and glutamate, which are essential for energy reserves, amino acid mobilization, and nitrogen transport [30,46]. These observations point to a more favorable uterine environment in multiparous cows, enabling enhanced nutrient availability during fetal development. This is in line with previous studies using the same animals [26] which showed that heifers born to multiparous cows were heavier at 13−15 months age, presented a higher body condition score, greater carcass fat deposition, and lower serum IGF-1 concentration compared to heifers from SEC cows, reflecting a better metabolic status that supports earlier puberty and reproduction readiness. Nishimura et al. [25] also reported that heifers from multiparous cows presented a higher number of cumulus-oocyte complexes, which were heavier than those collected from heifers born to second-parity cows. Pathway analysis in the present study further supports these observations, with significant differences identified in nicotinate and nicotinamide metabolism, phenylalanine and tyrosine metabolism, and pyrimidine metabolism, among others. These pathways, critical for energy production, amino acid utilization, and cellular growth, highlight the developmental advantages of heifers born to multiparous cows, whereas the metabolic challenges in second-parity cows likely reduce nutrient availability for the developing fetus [47–49].

When comparing the metabotype of heifers considering the weaning strategy, it was observed that the EW group exhibited significantly higher concentrations of valine, alanine, lactate, methionine, arginine, hippurate, isoleucine, and creatine phosphate compared to the CW group. These metabolites are involved in key metabolic pathways enriched in this study, including urea cycle, branched-chain amino acid (BCAA) metabolism, gluconeogenesis, pyruvate metabolism, arginine and proline metabolism, and glutathione metabolism. Notably, similar metabolic pathways, such as the glucose-alanine cycle, urea cycle, and glutathione metabolism, were enriched in Nellore heifers receiving creep-feeding supplementation, which also improved their reproductive performance [50]. This alignment underscores the critical metabolic shifts induced by enhanced nutrient availability, whether through the metabolic programming observed in the present study or the supplementation provided by creep-feeding. These findings highlight the vital role of early nutritional interventions in enriching essential metabolic pathways, ultimately promoting growth, metabolic adaptation, and improved reproductive outcomes.

The elevated concentrations of valine and isoleucine, two essential BCAAs, are particularly noteworthy due to their roles in rumen microbial protein synthesis – accounting for up to 90% of amino acids reaching the small intestine in cattle [51] – and as key substrates for energy production under metabolic stress. This may reflect enhanced mobilization of amino acids to fuel gluconeogenesis and energy metabolism for EW heifers. The increase in alanine further underscores these shifts in amino acid and energy metabolism, as alanine is synthesized from pyruvate and BCAAs in muscle tissue and accounts for a significant proportion of amino acid nitrogen transported to the liver in ruminants, playing a central role in nitrogen transport and gluconeogenesis [52].

Similarly, the higher levels of lactate in the EW group align with enhanced pyruvate metabolism and gluconeogenesis, indicating a shift toward anaerobic glycolysis. Lactate, produced as a by-product of pyruvate under anaerobic conditions, serves as a crucial gluconeogenic substrate in the liver [53]. The observed increase in creatine phosphate provides additional evidence of enhanced muscle energy metabolism in programmed heifers. As a rapid reservoir of high-energy phosphates, creatine phosphate plays a pivotal role in regenerating ATP during periods of high energy demand, particularly through the action of creatine kinase [54]. The higher concentrations of arginine and methionine in the EW group are also significant in the context of altered amino acid metabolism. Arginine is central to urea cycle function and serves as a precursor for nitric oxide synthesis, which is critical for vascular regulation and immune responses [55]. Meanwhile, methionine plays a key role in glutathione metabolism and methylation reactions through its derivative, S-adenosylmethionine [56].

Overall, these data may suggest an increased dependence on amino acids and glycolytic intermediates to maintain glucose homeostasis for programed heifers, thereby supporting energy requirements under conditions of altered nutrient partitioning. Additionally, the EW group may exhibit an upregulation of antioxidant defense mechanisms and detoxification processes, which are especially important during periods of oxidative stress.

Despite these significant metabolic differences, reproductive performance, body development, and hepatic gene expression in these animals were not notably affected in a parallel study [26]. This lack of phenotypic impact may be attributed to the optimal nutritional conditions provided during the study, which likely mitigated the physiological consequences of EW. However, in environments where forage quality or feed nutritive value is limited during mid-to-late lactation, these metabolic differences may result into more pronounced phenotypic effects, particularly in younger, still-growing cows [28].

Several studies have demonstrated that lactation significantly increases maintenance energy requirements in cows [57–59]. In this context, EW can lead to a notable reduction in the cow's maintenance energy expenditure by terminating milk production and shifting nutrient priorities. The energy conserved from reduced maternal maintenance and lactation demands can be redirected to enhance calf growth, either through additional feed intake or improved nutrient utilization efficiency [60]. Our findings suggest that EW is particularly advantageous in nutrient-limited systems, where it mitigates the trade-off faced by lactating cows between sustaining milk production and supporting fetal or post-weaning growth. This improves metabolic efficiency and enhance system sustainability by reallocating energy toward fetal development.

Therefore, EW strategies should be assessed within the framework of environmental and nutritional conditions, as their benefits may be more pronounced under circumstances of limited feed availability or elevated maternal metabolic and energy demands.

## 5. Conclusion

This study demonstrated that the metabotype of heifers is influenced by fetal programming through weaning strategy, as well as by maternal parity, with early weaning exerting a greater impact on heifers born to second-parity cows due to their dams' ongoing growth requirments. The observed interaction between parity and weaning strategy underscores the heightened metabolic challenges faced by these heifers. Our novel findings emphasize the overall benefits of early weaning as a fetal programming strategy to enhance nutrient utilization efficiency, particularly under challenging conditions such as limited feed availability or periods of elevated physiological demand in the dam. In conclusion, early weaning at 5 months of age can be recommended across all cows categories; however, second-parity cows and those that conceived late in the breeding season may derive greater benefit from this strategy. This is attributable to the continued maternal growth in second-parity cows and the limited recovery period for body condition in late-conceiving cows before the subsequent breeding season. Furthermore, cows with lower body condition scores may also benefit from early weaning. Importantly, the success of this management approach depends on ensuring adequate post-weaning nutrition for calves weaned at 5 months of age.

## Supporting information

**Supplementary Table S1. Assignments of the $^1$H NMR spectrum of the metabolites identified in the serum sample in 0.1 mol L$^{-1}$ phosphate buffer. Chemical shifts (in ppm). multiplicity. coupling constants (in Hz) for hydrogens.**
(DOCX)

**Supplementary Table S2. Metabolites concentration in heifers' serum that didn't presented the effect of interaction Parit*Treat.**
(DOCX)

**Supplementary Table S3. Metabolites concentration per animal (all data available).**
(XLSX)

**Supplementary Figure S1. Combined power plot.**
(PNG)

## Author contributions

**Conceptualization:** Ana Laura dos Santos Munhoz Gôngora, Thiago Kan Nishimura, Tarique Hussain, Arlindo Saran Netto, Guilherme Pugliesi, Nara Regina Brandão Cônsolo.

**Data curation:** Thiago Kan Nishimura, Gabriel Henrique Ribeiro, Eduardo Solano Pina Santos.

**Formal analysis:** Gabriel Henrique Ribeiro, Eduardo Solano Pina Santos, Luiz Alberto Colnago.

**Funding acquisition:** Paulo Roberto Leme, Guilherme Pugliesi, Nara Regina Brandão Cônsolo.

**Investigation:** Ana Laura dos Santos Munhoz Gôngora, Thiago Kan Nishimura, Alanne Tenório Nunes, Amjad Hameed, Luiz Alberto Colnago, Paulo Roberto Leme, Arlindo Saran Netto, Guilherme Pugliesi, Nara Regina Brandão Cônsolo.

**Methodology:** Thiago Kan Nishimura, Gabriel Henrique Ribeiro, Eduardo Solano Pina Santos, Alanne Tenório Nunes, Tarique Hussain, Luiz Alberto Colnago, Nara Regina Brandão Cônsolo.

**Project administration:** Paulo Roberto Leme, Arlindo Saran Netto, Guilherme Pugliesi, Nara Regina Brandão Cônsolo.

**Resources:** Paulo Roberto Leme, Guilherme Pugliesi.

**Supervision:** Paulo Roberto Leme, Guilherme Pugliesi, Nara Regina Brandão Cônsolo.

**Validation:** Luiz Alberto Colnago.

**Visualization:** Anjaleena Yaseen, Ana Laura dos Santos Munhoz Gôngora, Eduardo Solano Pina Santos, Amjad Hameed, Luiz Alberto Colnago.

**Writing – original draft:** Anjaleena Yaseen, Alanne Tenório Nunes.

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
