## [Decision Letter · Decision Letter 0]

Dear Dr. Cônsolo,

Thank you for submitting your manuscript to PLOS ONE. After careful consideration, we feel that it has merit but does not fully meet PLOS ONE’s publication criteria as it currently stands. Therefore, we invite you to submit a revised version of the manuscript that addresses the points raised during the review process.

Considering the reviewers' assessments, I request that the authors accept the suggestions and return the paper so that we can proceed. 

Please make the highlighted adjustments so that the necessary adjustments can be observed. 

Att 

We look forward to receiving your revised manuscript.

Kind regards,

Julio Cesar de Souza, Ph.D.

Academic Editor

PLOS ONE

2. We note that this submission includes NMR spectroscopy data. We would recommend that you include the following information in your methods section or as Supporting Information files:

1) The make/source of the NMR instrument used in your study, as well as the magnetic field strength. For each individual experiment, please also list: the nucleus being measured; the sample concentration; the solvent in which the sample is dissolved and if solvent signal suppression was used; the reference standard and the temperature.

2) A list of the chemical shifts for all compounds characterised by NMR spectroscopy, specifying, where relevant: the chemical shift (δ), the multiplicity and the coupling constants (in Hz), for the appropriate nuclei used for assignment.

3)The full integrated NMR spectrum, clearly labelled with the compound name and chemical structure.

We also strongly encourage authors to provide primary NMR data files, in particular for new compounds which have not been characterised in the existing literature. Authors should provide the acquisition data, FID files and processing parameters for each experiment, clearly labelled with the compound name and identifier, as well as a structure file for each provided dataset. See our list of recommended repositories here: https://journals.plos.org/plosone/s/recommended-repositories "

 [This study was supported by The Coordenação de Aperfeiçoamento de Pessoal de Nível Superior - Brasil (CAPES) - Finance Codes 001, and Fundação de Amparo à Pesquisa do Estado de São Paulo (FAPESP) grant 2020/08845-3, 2017/18937-0, 2022/07906-4.]. 

5. In the online submission form, you indicated that [Data can be shared once required to the corresponding author].

This policy applies to all data except where public deposition would breach compliance with the protocol approved by your research ethics board. If your data cannot be made publicly available for ethical or legal reasons (e.g., public availability would compromise patient privacy), please explain your reasons on resubmission and your exemption request will be escalated for approval. "

Reviewers' comments:

Reviewer's Responses to Questions

**Comments to the Author**

1. Is the manuscript technically sound, and do the data support the conclusions?

Reviewer #1: Yes

Reviewer #2: Partly

Reviewer #3: Yes

Reviewer #4: Yes

2. Has the statistical analysis been performed appropriately and rigorously?

Reviewer #1: I Don't Know

Reviewer #2: Yes

Reviewer #3: Yes

Reviewer #4: Yes

3. Have the authors made all data underlying the findings in their manuscript fully available?

Reviewer #1: Yes

Reviewer #2: No

Reviewer #3: Yes

Reviewer #4: Yes

4. Is the manuscript presented in an intelligible fashion and written in standard English?

Reviewer #1: No

Reviewer #2: Yes

Reviewer #3: Yes

Reviewer #4: Yes

Reviewer #1: Fetal Programming through Early Weaning Shapes the Metabotype of Nelore Heifers is very interesting study for animal nutritionist. However, authors Used “and” and “metabolism “extensively throughout the text. Need to improve sentences. Generally, need to improve English. I will suggest few changes in the manuscript before publication. Highlighted the words need to rewrite. These changes are following: Write full spellings of abbreviations used in abstract.

On line #n 29: Write full spelling of PLS-DA when you are using first time in the text. Please, also consider it for all abbreviations if there is any same case.

On line #37: Please, mention plasma or any other organ parameters such as phenylalanine, creatine phosphate, and glutamines?

On line # 37: what nutrients availability?

What authors will suggest to the animal nutritionist for feed formulation or nutrients requirements for better development considering parity and weaning from this study?

Do authors think metabotype results will have same findings in cows weaned on other forages spp rather than Brachiaria spp. Pasture as in this study.

Secondly, can we apply these results on for this cow Nellore breed or for other cow breeds as well as for sheep/goat etc.

How these results can improve meat quality?

Reviewer #2: This study investigates the effects of early weaning (EW) on the metabolic programming of Nelore heifers, with a focus on interactions between weaning strategy and dam parity (multiparous vs. secundiparous). The research design is logical, and the data provide valuable insights into optimizing cattle management strategies in tropical regions. The manuscript is well-structured, and the methodology is generally sound. However, certain methodological and interpretative aspects require clarification and refinement to enhance scientific rigor.

Major comments:

1. The manuscript lacks data on feed quality and nutrient intake during the dry season. Including details on feed composition and energy intake would clarify whether "nutritional stress" was experimentally relevant.

2. A statistical power analysis or effect size estimation should be included to validate the robustness of group comparisons.

3. The study focuses on fetal programming effects induced by early weaning, yet the serum sampling was conducted at 14 months of age rather than immediately post-birth. The authors should clarify the rationale for selecting this specific time point.

Minor comments:

1. There are a few minor grammatical errors and typos that should be corrected to polish the text. For instance, in one sentence: “heifers born to multiparous cows had were heavier from 13 to 15 months age…” – this phrasing is incorrect and can be simplified (perhaps “heifers born to multiparous cows were heavier at 13–15 months of age…”). Similarly, “this data my suggests” should likely be “this data may suggest”, and “gene expresison” should be “gene expression”.

2. Line 58 and 59: the citation style here need to be consistent with other places. These references are also missing in the list.

3. Standardize terms (e.g., "metabotype" vs. "metabolic profile") throughout the text.

4. References 23–24 are listed as "Not Published." Replace with formal citations (e.g., DOI links for preprints) or clarify their status.

5. Aside from Table 1, if the authors have additional data (all 39 metabolites measured), providing a full table (perhaps as a supplement) could be beneficial for transparency. This would let interested readers see the magnitude of differences even for non-significant metabolites.

Reviewer #3: Thank you for this work. It contains many interesting approaches, especially for beef farmers in nutrient-poor areas, for managing cows optimally and ensuring the good development of their unborn calves. I am sure that these approaches will help to optimize productivity in beef production. I have only a few comments and hope to make the paper stronger with them.

In the discussion, you mentioned the metabolites in part. Can you go into that in more detail? Why did you decide to examine these metabolites in advance? I think, it would make it easier for the reader to understand the selection of the metabolites and enables a better understanding of the publication without having to conduct additional research.

Would you generally recommend early weaning at 5 months for beef cows, or does that only apply to cows calving for the first and second time? What consequences would these results have for the field?

Were blood samples also taken from the cows to examine the metabolites? I can imagine that there are also differences between the cows, depending on their parity. What about the animal's individual constitution? Do you think that this also has an influence on fetal programming?

A non-publication related question, based on interest: Are there differences in the success of artificial insemination in cows with calves that were weaned earlier? Will this be presented in further publications? Here, too, it may be expected that there would be a difference between the individual groups (EW, CW) and the parity of the dam. Considering that primiparious cows need longer to recover from calving and used their energy for there own growing, could this also be a consequence of later insemination in the next breeding season? This would also be important information for practitioners to improve the productivity in beef cattle production. I think it would be very important to address this issue in a follow-up paper.

Reviewer #4: Fetal Programming through Early Weaning Shapes the Metabotype of Nelore Heifers

General Assessment

This manuscript presents a well-designed study that explores the impact of fetal programming, induced by early weaning of the previous calf, on the metabolic profile (metabotype) of Nelore heifers. The research addresses a relevant and underexplored question in beef cattle production systems, particularly in tropical environments where nutritional challenges prevail during the dry season. The use of ¹H-NMR metabolomics combined with a factorial experimental design provides robust insights into how early maternal management strategies can influence offspring physiology. The study is clearly written, methodologically sound, and provides novel data that may be valuable for improving cow-calf management and reproductive efficiency in Bos indicus systems.

Nonetheless, some revisions are required before the manuscript can be considered for publication. These revisions include improving the clarity and precision of specific terminology, ensuring compliance with PLOS ONE policies regarding data and funding disclosures, and strengthening the discussion on the biological relevance of key metabolites.

Major Comments

**Terminology Consistency – Use of "Secundiparous"**

The manuscript frequently uses the term “secundiparous,” which is uncommon and may cause confusion. It is recommended to replace this term throughout the manuscript with “second-parity” or “second-calving” cows, which are more standard in the scientific literature.

• **Hypothesis Statement**

While the hypothesis is presented at the end of the introduction, it would benefit from a more direct formulation. A suggested revision: “We hypothesize that early weaning of the previous calf programs the metabolic profile of heifers differently depending on dam parity.”

• **Interpretation of Biological Relevance**

Although key metabolites (e.g., glutamate, acetate, hippurate) are statistically different, their specific roles in pubertal development or reproductive physiology are not always clearly linked in the discussion. The authors are encouraged to expand on the physiological implications of these differences, particularly how they might translate into functional outcomes such as fertility, immunity, or growth performance under commercial conditions.

• **Conclusion Enhancement**

The conclusions are well supported by the data but could be expanded to include practical implications. For example, how might early weaning strategies be integrated into cow-calf management programs under tropical forage systems?

Minor Comments

• **Table 1 Formatting**

Add a full descriptive title and define the superscript letters (“a”, “b”) clearly in the footnote. Units (mmol) should also be clearly labeled in the column headers.

• **Figures**

Ensure all figures (1–6) are provided in high resolution with stand-alone legends. Each figure should be interpretable without reference to the main text, as per PLOS ONE’s guidelines.

Compliance with Journal Policies

• **Financial Disclosure Statement**

The authors must revise the financial disclosure to align with PLOS ONE policy. Specifically, identify which authors received which grants, and include URLs to the funding agencies. Also, confirm whether the funders had any role in the study design or data interpretation.

Final Recommendation

**Recommendation**

This is an important and well-executed study that offers meaningful insights into fetal programming in beef cattle. After addressing the points above, especially the mandatory compliance items related to journal policy, the manuscript will be suitable for publication.

**Do you want your identity to be public for this peer review?** For information about this choice, including consent withdrawal, please see our Privacy Policy

Reviewer #1: No

Reviewer #2: **Yes: ** Yao Xiao

Reviewer #3: **Yes: ** Dr. Linda Dachrodt

Reviewer #4: No

---

## [Author Response · Author response to Decision Letter 1]

4 Jun 2025

We included a reviewers answer file

---

## [Editor Report · Decision Letter 1]

Fetal Programming through Early Weaning Shapes the Metabotype of Nelore Heifers

PONE-D-25-05801R1

Dear Dr. Cônsolo,

We’re pleased to inform you that your manuscript has been judged scientifically suitable for publication and will be formally accepted for publication once it meets all outstanding technical requirements.

Kind regards,

Julio Cesar de Souza, Ph.D.

Academic Editor

PLOS ONE

Additional Editor Comments (optional):

Considering the suggestions of the reviewers and the adjustments made by the authors, I am in favor of publishing it.

Best regards,

JCS
---

## [Editor Report · Acceptance letter]

PONE-D-25-05801R1

PLOS ONE

Dear Dr. Cônsolo,

I'm pleased to inform you that your manuscript has been deemed suitable for publication in PLOS ONE. Congratulations! Your manuscript is now being handed over to our production team.

Kind regards,

on behalf of

Dr. Julio Cesar de Souza

Academic Editor

PLOS ONE